# Effect of Alloying Elements on the High-Temperature Yielding Behavior of Multicomponent γ′-L1_2_ Alloys

**DOI:** 10.3390/ma17102280

**Published:** 2024-05-11

**Authors:** Chen-Yuan Wang, Sae Matsunaga, Yoshiaki Toda, Hideyuki Murakami, An-Chou Yeh, Yoko Yamabe-Mitarai

**Affiliations:** 1Department of Advanced Materials Science, The University of Tokyo, 5-1-5 Kashiwanoha, Kashiwa-shi 277-8561, Chiba, Japan; 1775090211@edu.k.u-tokyo.ac.jp; 2National Institute for Materials Science, 1-2-1 Sengen, Tsukuba 305-0047, Ibaraki, Japan; toda.yoshiaki@nims.go.jp (Y.T.); murakami.hideyuki@nims.go.jp (H.M.); 3Department of Nanoengineering and Nanoscience, Waseda University, Shinjuku 169-8555, Tokyo, Japan; 4High Entropy Materials Center, National Tsing Hua University, Hsinchu 30013, Taiwan; yehac@mx.nthu.edu.tw

**Keywords:** multicomponent alloy, high-temperature deformation, intermetallic compounds, CALPHAD

## Abstract

The exceptional mechanical properties of Ni-based high entropy alloys are due to the presence of ordered L1_2_ (γ′) precipitates embedded within a disordered matrix phase. While the strengthening contribution of the γ′ phase is generally accepted, there is no consensus on the precise contribution of the individual strengthening mechanisms to the overall strength. In addition, changes in alloy composition influence several different mechanisms, making the assessment of alloying conditions complex. Multicomponent L1_2_-ordered single-phase alloys were systematically developed with the aid of CALPHAD thermodynamic calculations. The alloying elements Co, Cr, Ti, and Nb were chosen to complexify the Ni_3_Al structure. The existence of the γ′ single phase was validated by microstructure characterization and phase identification. A high-temperature compression test from 500 °C to 1000 °C revealed a positive temperature dependence of strength before reaching the peak strength in the studied alloys NiCoCrAl, NiCoCrAlTi, and NiCoCrAlNb. Ti and Nb alloying addition significantly enhanced the high-temperature yield strengths before the peak temperature. The yield strength was modeled by summing the individual effects of solid solution strengthening, grain boundary strengthening, order strengthening, and cross-slip-induced strengthening. Cross-slip-induced strengthening was shown to be the key contributor to the high-temperature strength enhancement.

## 1. Introduction

Ni-based superalloys have been extensively employed as critical high-temperature structural materials used in aerospace engineering and the power generation industry due to their superior high-temperature mechanical properties, which are derived from the combination of 60–70% volume fraction of ordered γ′ precipitates (L1_2_) coherently embedded in the solid-solution γ matrix (FCC-A1) [1]. The alloy design of Ni-based superalloys has approached its limit for performance enhancements due to the constraints of Ni-based systems. A novel concept in alloy design, high entropy alloys (HEAs), has made a breakthrough in conventional alloys by revisiting the vast composition space [2]. Precipitation-strengthened HEAs, as newly emerging structural materials, are highly attractive due to their combination of strength and ductility [3,4,5,6,7,8,9]. Compared to grain boundary strengthening and solid solution strengthening, precipitation strengthening of multicomponent L1_2_ nanoprecipitates can significantly improve the yield strength of HEAs while maintaining good ductility. Yeh et al. [9] first introduced the L1_2_ nanoprecipitates of (Ni, Co)_3_Ti into the FeCoNiCr system by adding Ti to form a precipitation-strengthened HEA. Later, Yang et al. [8] designed and developed a complex alloy (FeNiCo)_86_Al_7_Ti_7_ with multicomponent L1_2_ intermetallic particles (Ni, Co, Fe)_3_(Ti, Al, Fe) and reported a combination of high strength (~1 GPa) and excellent ductility (~50%) at room temperature. They attributed the pronounced increase in yield strength to the high ordering strengthening by the high density of L1_2_ nanoprecipitates and the higher anti-phase boundary energy (APBE) provided by Ti addition.

Recently, Yang et al. [10] developed a multicomponent L1_2_-ordered alloy, NiCoFeAlTiB, and observed that the materials did not experience obvious softening behavior below 800 °C under the hardness test. Long et al. [11] investigated a multicomponent Co-based L1_2_ ordered single phase intermetallic alloy and found that the alloy exhibited anomalous yield strength increase from 250 °C to 800 °C. The stress anomaly was stronger than ternary L1_2_ Co_3_(Al, W) and B-doped Ni_3_Al. These results revealed that a high-temperature capability could be achieved by the complex L1_2_ intermetallic. It is well known that the L1_2_ phase exhibits anomalous yielding behavior, represented by a substantial yield strength increase at intermediate temperatures [1]. This positive temperature dependence of stress of L1_2_ precipitates confers excellent high-temperature mechanical properties. It has been generally accepted that the anomalous yielding behavior was due to the Kear–Wilsdorf mechanism, forming anti-phase boundary (APB)-coupled dislocation pairs by thermally activated cross-slip of a screw segment of the superpartial dislocation from octahedral {111} planes to cube {010} planes. While the strength in Ni-based superalloys is proportional to the APBE, the anisotropy in APBE between octahedral {111} planes and cube {010} planes determines the yield strength anomaly. Since a perfect dislocation slipping on the plane in Ni_3_Al consists of four Shockley partial dislocations and bounding one APB fault and two complex stacking faults (CSF), the occurrence of thermally activated cross-slip process is dependent on APB and CSF energies [12,13].

Considerable efforts have been devoted to researching the composition dependence of the high-temperature strength and associated planar fault energies [14,15,16,17,18]. Mishima et al. [14,19] systematically investigated the effect of ternary addition on the temperature dependence of strength in ternary Ni_3_Al compounds. Among the investigated alloying elements, Ti, Zr, Hf, V, Nb, Ta, Mo, W, Ti, Nb, and Ta produced the most drastic strength increment. Moreover, advances in computational models and modern computers allow for the APBE of the ternary Ni_3_Al intermetallic compounds to be predicted by density function theory (DFT) calculations [13,20,21,22,23,24]. Gorbatov et al. [23] studied the effect of composition on the APBE on both octahedral planes and cube planes of Ni-based L1_2_-ordered alloys by ab initio calculations and reported ternary alloying additions. Ti, V, and Cr occupied the Al sublattice site and increased the APBE for both {111} and {010} planes, while Co, Cu, and Fe occupied the Ni sublattice site and sightly affected both the {111} and {010} APBE.

Recent studies on the deformation mechanisms of L1_2_-strengthened alloys revealed that γ′-strengtheners, such as Ti, Ta, and Nb, significantly improved the creep performance by forming the other phases at planar defects, which are considered to have slower deformation kinetics [25]. In addition, the experimental and computational works showed Co and Cr and γ′ strengtheners significantly affected the phase transformation, which preferentially occurred at planar defects. While forming an ordered phase with slower formation kinetics can enhance the high-temperature creep performance, the formation of disordered phases will degrade it. Determining the resulting phase hinges on subtle discrepancies in nominal alloy composition [25,26,27,28]. Although the deformation mechanisms of γ′-strengthened alloys have been investigated for decades, there are still some uncovered questions about the γ′ phase. In γ + γ′ structures, deformation mechanisms are usually significantly affected by the interactions at γ and γ′ interface [29]; thus, elucidating the strengthening effect solely attributable to the γ′ phase has proven to be a formidable task. Furthermore, while numerous recent studies have been dedicated to the effect of alloying elements on the γ′-shearing mechanisms, their influence on phenomena unique to the L1_2_ structure, such as the elevation of yielding stress at high temperatures, remains significantly underexplored in a systematic manner.

Fundamental and systematic studies on multicomponent L1_2_ phases are urgently required to accelerate the advancement of L1_2_-strengthened alloys. The precipitation strengthening by L1_2_-type Ni_3_Al nanoparticles has been extensively investigated in Ni-based superalloys and HEAs. However, most previously investigated Ni_3_Al phases are compositionally simple and contain a lower level of ternary elements. Because of the extensive alloying addition of Cr and Co in Ni-based superalloys and L1_2_ precipitation strengthened HEAs and their effect on local phase transformation, these two elements were chosen as a base with Ni_3_Al to design multicomponent L1_2_ alloys. This work systematically varied compositions by substituting Ni and Al with Ti or Nb to study the compositional effects on the yielding behaviors of the L1_2_ phase. The investigated multicomponent L1_2_ alloys were systematically designed using the thermodynamics software CALPHAD Thermo-Calc TCNI8 Ni-based Superalloys Database [30]. The yielding behaviors of the multicomponent L1_2_ alloys were evaluated by high-temperature compression tests. The analysis and discussion will delve into the contribution of different strengthening mechanisms under various alloying conditions.

## 2. Materials and Methods

The multicomponent γ′ (L1_2_-ordered) single phase was designed by utilizing the Thermo-Calc© TCNI8 database [30], aiming to obtain L1_2_-ordered single phase within the temperature range between 800 °C and 950 °C. We developed three systems, as follows:NiCoCrAl,NiCoCrAlTi,NiCoCrAlNb.

The pseudo-binary isopleths and volume fraction diagrams of three multicomponent systems were calculated to confirm the solubility limit of Ti or Nb alloying addition in NiCoCrAl alloy without forming any secondary phases. Ti and Nb were carefully added to avoid exceeding their solubility limits, approximately 2 at%, in the NiCoCrAl alloy. Selected compositions for the studied alloys are listed in Table 1 and marked using the dash lines in Figure 1.

In total, 300 g of the studied alloy ingots were prepared by the vacuum arc melting (VAM) process under an argon atmosphere with a Ti getter. Repeated melting was carried out at least six times to ensure the chemical homogeneity. As-cast samples were homogenized at 980 °C for 168 h, followed by air cooling to room temperature. For CY3 alloy with Nb alloying addition, a prior solution heat treatment (SHT) at 1280 °C for 168 h was necessary to eliminate the dendritic structure caused by the severe Nb segregation. The constituent phases were identified using an X-ray diffractometer (XRD, SmartLab, Rigaku, Tokyo, Japan). The microstructures of each alloy were characterized using scanning electron microscopy (SEM, JSM-7200 F, JEOL, Tokyo, Japan). The phase transformation temperatures of the studied alloys were determined by differential thermal analysis (DTA, Labsys DSC/DTA, Setaram, Caluire-et-Cuire, France) experiments between 25 °C and 1500 °C, with a scanning rate of 10 °C/min. Cuboid compression specimens with 3 mm and 6 mm width and height length were sectioned by a high-speed precision cutting machine. High-temperature compression tests were performed to understand the strength evolution at 500, 600, 700, 800, 900, and 1000 °C with a strain rate of 10^−4^ s^−1^. All the tests were interrupted after reaching approximately 10% plastic strain.

## 3. Results and Discussion

### 3.1. Microstructures

The volume fraction diagrams of the designed alloys are shown in Figure 2. The γ’ (L1_2_-ordered) single phase exists in the range of 659–1104 °C in CY1, 870–1060 °C in CY2, and 858–991 °C in CY3. The temperature window with only the L1_2_-ordered single phase narrowed by adding the fifth element, while the γ + γ’ two-phase window broadened. The γ + γ′ window is 1104–1163 °C in CY1, 1060–1171 °C in CY2, and 991–1133 °C in CY3. Both Ti and Nb were classified as γ′-stabilizers in terms of the alloy design strategy for γ + γ′ two-phase Ni-based superalloys [1]. Our calculation results indicated that these elements stabilized the L1_2_-ordered phase in a wide range of temperatures with the existence of the γ matrix while they decreased the L1_2_-ordered single-phase window. We also investigated the site occupancy in each alloy using thermodynamic calculations. It was simulated that Co tended to occupy the Ni sublattice sites, while Cr, Ti, and Nb atoms substituted the Al site atoms to create (Ni, Co)_3_(Al, Cr) in CY1, (Ni, Co)_3_(Al, Cr, Ti) in CY2, and (Ni, Co)_3_(Al, Cr, Nb) in CY3.

Figure 3 shows the microstructure of the CY1, CY2, and CY3 alloys after homogenizing heat treatment. All alloys exhibited single-phase microstructures after the heat treatments. The average grain size of each alloy is around 503 μm in CY1, 849 μm in CY2, and 323 μm in CY3. Any secondary phases were not detected either within the grains or at the grain boundaries, as shown in Figure 3d–f, which confirms that we successfully designed and manufactured the multicomponent L1_2_-ordered single-phase alloys.

Figure 4 shows the indexed XRD spectra peaks. The 2θ values, corresponding to the indexed diffraction peaks of the studied alloys, are listed with the ones of Ni_3_Al in JCPDS files in Table 2. The 2θ values of the studied alloys CY1, CY2, and CY3 agreed well with the values in JCPDS files, which confirmed the existence of an L1_2_-ordered single phase in the studied alloys. Minor differences from the JCPDS files could arise from the peak broadening due to differences in grain size, lattice strain, and changes in lattice parameters [31]. The crystal structure and microstructure of the alloys after the heat treatments were in good agreement with the prediction from the calculated equilibrium phase diagrams.

Figure 5 shows the DTA heating curves of the heat-treated CY1, CY2, and CY3 alloys. The onset temperature of endothermic peaks was determined as the phase transformation temperatures and solvus temperatures. The phase transformation temperatures (PTTs) were defined as the temperatures at which the alloys could retain the L1_2_-ordered single phase. Above these temperatures, the secondary and third phases started to form. The PTTs and solvus temperatures are summarized in Table 3. The PTTs of the CY1, CY2 and CY3 alloys were 1217 °C, 1244 °C, and 1298 °C, respectively. The results showed that the L1_2_-ordered single phase persisted at higher temperatures in the quinary alloys with Ti or Nb alloying additions. In the Ni-based superalloys with γ + γ′ structure, adding Ti or Nb increased γ′-solvus temperatures, indicating these elements could stabilize the L1_2_-ordered phase at elevated temperatures. Since our alloys were designed only to have an L1_2_-ordered single phase, the aforementioned effect of γ′-formers was observed as the expansion of an L1_2_-ordered single-phase regime. The solidus temperatures for the CY1, CY2, and CY3 alloys were 1353 °C, 1324 °C, and 1305 °C, respectively. The decrease in solidus caused by Ti alloying addition was also observed in Ni-based superalloys due to the strong partition tendency of Ti to liquid phase, forming segregation phases with lower melting temperatures and reducing the solidus temperature of the alloy [32,33].

### 3.2. Yielding Behavior through Temperatures

Figure 6 depicts the 0.2% yield stress measured from the high-temperature compression tests for CY1, CY2, and CY3 along the deformation temperatures. The yield stress measured at each temperature is summarized in Table 4. Regardless of the testing temperature, CY3 exhibited the highest yield stress among the three alloys before the peak temperature. All alloys exhibited yield stress increasing monotonically before reaching a maximum and then significantly decreasing with increasing the testing temperature after reaching the peak temperature. The difference in the yield stresses significantly decreased after the peak temperatures.

Comparing the yield stress of CY1, the ones of CY2 are 30% higher at 500 °C, 31% higher at 600 °C, and 59% higher at 700 °C. In addition, the yield stresses of CY3 are 81% higher at 500 °C, 68% higher at 600 °C, and 62% higher at 700 °C. The peak strengths for CY1, CY2, and CY3 alloys were 426 MPa, 678 MPa, and 690 MPa, respectively. Our results revealed that the fifth alloying addition produced a significant enhancement in the yield stress while exhibiting anomalous yielding behavior, which was reported in numerous studies about the high-temperature mechanical behavior of L1_2_-Ni_3_Al. Mishima et al. [14] investigated the effect of ternary addition of several fourth, fifth, and sixth group elements on the high-temperature mechanical response of polycrystalline Ni_3_Al and reported that 4 at% Ti addition, which substituted at Al sites, reached the peak strength of about 560 MPa and 2 at% Nb substitution obtained the peak strength of about 620 MPa at 600 °C. It has been generally accepted that Co and Cr addition in Ni-based superalloys strengthen the γ matrix [1]. However, in this research, comparing the compression results of polycrystalline Ni_3_Al from Lopez and Hancock [17] with CY2 and CY3, a noticeable strength enhancement was obtained by adding Co and Cr in Ni_3_Al. Furthermore, comparing the three alloys studied in the research, the Ti or Nb alloying addition showed a significant increase in peak strength.

During the high-temperature compression deformation up to the peak temperature, the yield strength of the alloys depends on the solid solution strengthening (${∆\sigma }_{SS}$), grain boundary strengthening by the Hall–Petch effect (${∆\sigma }_{gb}$), ordering strengthening (${∆\sigma }_{OS}$), and cross-slip-induced strengthening (${∆\sigma }_{KW}$) [34,35,36], as follows:(1)$$\sigma_{y}={∆\sigma }_{SS}+{∆\sigma }_{gb}+{∆\sigma }_{OS}+{∆\sigma }_{KW}.$$

We evaluated the contribution of each strengthening mechanism at peak temperature (800 °C for CY1 and 700 °C for CY2 and CY3) using Equation (1) above. The contribution of solid solution strengthening, ${∆\sigma }_{SS}$, was calculated using the pySSpredict, a Python-based toolkit that automates the high-throughput calculations of solid solution strength of complex concentrated alloys based on the solid solution strengthening and edge dislocation models for FCC and BCC alloys [37,38]. The calculated values of ${∆\sigma }_{SS}$ are 20 MPa, 24 MPa, and 32 MPa for the CY1, CY2, and CY3 alloys, respectively. Nb produced the most prominent strength enhancement via the solid solution strengthening, followed by Ti addition, which was consistent with the measurement of lattice constants through XRD.

The grain boundary strengthening effect can be expressed as the Hall–Petch relationship [39]:(2)$${∆\sigma }_{gb}=\sigma_{0}+kd^{-1},$$
where $\sigma_{0}$ is the lattice friction stress, k is the Hall–Petch coefficient, and d is the average grain size. Since the average grain size of our alloy is ≈500 μm for CY1, ≈850 μm for CY2, and ≈330 μm for CY3, the effect of grain boundary strengthening is expected to be relatively small in all alloys.

In addition to the solid solution strengthening and grain boundary strengthening, the planar fault energy was assumed to contribute to the significant strength enhancement [40]. The strengthening effect of the L1_2_ phase is mainly due to the ordering strengthening caused by adding different elements. Since the bonding force between atoms of different elements is greater than between atoms of the same elements, the ordered arrangement of atoms of different types will contribute to a higher strength for the ordered alloy, according to the expression of the ordering strengthening (${∆\sigma }_{OS}$) [41]:(3)$${∆\sigma }_{OS}=M0.81\frac{\gamma_{APB}}{2b}{\left(\frac{3\pi f}{8}\right)}^{1/2}\;,\;$$
where M is the Taylor factor, $\gamma_{APB}$ is APBE, b is the burgers vector, and f is the volume fraction of the L1_2_ phase. Since APBE dominates the γ′-shearing event by both weakly-coupled dislocations and strongly-coupled dislocations, higher APBE would enhance the overall strengthening of the L1_2_-ordered phase before the peak temperatures. Chandran and Sondhi [42] investigated the effect of Ti and Nb on the Ni_3_Al by DFT calculations. They reported that Nb and Ti could significantly increase APBE but the Nb effect is stronger with Ni_3_Al_1−*x*_Nb*_x_* with *x*
≈ 0.20. This research used a DFT method, developed by Crudden et al. [22], to estimate the APBE of the studied alloys. It is assumed that the change in APBE can be determined using a linear superposition of the effects of the individual alloying elements according to the equation
(4)$$∆E_{APB}=E_{APB}^{0}+\sum_{i}^{n}\left(k_{i}x_{i}\right),$$
where $x_{i}$ is the concentration in at% of the solute element *i* in the L1_2_-ordered phase and $k_{i}$ is the coefficient for change in APBE, listed in Table 5. $E_{APB}^{0}$ is the APBE for Ni_3_Al (193 ± 13 mJ m^−2^) measured using TEM by Kruml et al. [43]. The APBE of the CY1, CY2, and CY3 alloys were 153 mJ/mol, 183 mJ/mol, and 196 mJ/mol, respectively. The calculated contribution values to yield strength from ordering strengthening, ${∆\sigma }_{OS}$, are 81 MPa, 97 MPa, and 104 MPa for CY1, CY2, and CY3, respectively. The calculation result indicates that adding Ti or Nb results in an obvious increase in the APBE and, thus, the contribution of ordering strengthening in the studied alloys.

With increasing temperature, the strength of the L1_2_-ordered phase increases, which is controlled by the thermally activated cross slip of dislocations from {111} to {010} planes. The number of dislocations in the L1_2_ phase increased as plastic deformation was induced, resulting in the difficulty in dislocation movement along the L1_2_ structure. This study estimated the contribution of cross-slip-induced strengthening (Δ*σ*_KW_) by subtracting the solid solution strengthening and the ordering strengthening from the total peak yield strength, summarized in Figure 7. The values obtained for cross-slip-induced strengthening are 325 MPa, 557 MPa, and 554 MPa for CY1, CY2, and CY3, respectively. Therefore, it was assumed that the cross-slip-induced strengthening dominated the peak yield strength and the significant strength enhancement was achieved by adding the fifth elements, namely Ti or Nb. Yu et al. [13] have investigated the effect of alloying element on dislocation in Ni_3_Al using the DFT method and reported that Ti addition could reduce the cross-slip activation enthalpy and facilitate the cross-slip process to form dislocation locks, thus resulting in more difficult dislocation movement and higher flow stress in the anomalous temperature regime of Ni_3_Al. According to our experimental results, Nb can provide a similar level of cross-slip-induced strengthening as Ti in the NiCoCrAl alloy system.

In addition to the strength enhancement, a slight decrease in the peak temperature for the positive temperature dependence was observed in the quinary CY2 and CY3 alloys. This phenomenon was also observed in previous studies with ternary alloying addition in Ni_3_Al alloys and the peak temperature would further decrease with increasing alloying addition of the ternary elements [14,16,17]. Lopez and Hancock [17] suggested that the decrease in peak temperature was due to the strong influence of Ti on the onset of cube slip. Kruml et al. [43] proposed and verified that increasing the CSF energy increased the strength up to the peak temperature and decreased the strength above the peak temperature, thus lowering the peak temperature. According to the CSF energies calculated by Yu et al. [13] using the DFT method, Ti was found to increase the CSF energy, consistent with the lower peak temperature observed in this research.

From a technical point of view, the peak temperature is usually at the highest temperature where the materials are used [44]. Ni-based superalloys [45,46,47,48] and L1_2_-strengthened HEAs [49,50,51,52] have exhibited anomalous positive temperature dependence of strength at intermediate temperatures followed by a decrease in yield strength with increasing temperatures. The decrease in strength has been explained mainly by degradation of microstructures and transition of deformation mechanisms. Since the L1_2_-ordered single phase microstructure in our systematically designed alloys was thermodynamically stable at the peak temperatures, the possibility of a microstructure degradation has been ruled out for the cause of the decreased yield strength above the peak temperatures. Our results revealed the intrinsic capability of the multicomponent L1_2_-ordered phases and suggested that the difference in the yield strength before the peak temperatures was significantly influenced by the difference in the contribution of cross-slip-induced strengthening, which was strongly associated with the alloying compositions of the multicomponent L1_2_-ordered phase. This work has shed light on the significance of cross-slip-induced strengthening in the multicomponent L1_2_-ordered phase and can serve as a guideline for the future design of L1_2_-strengthened HEAs.

According to the DFT calculation result of activation energy of the cross-slip process by Yu et al. [13], Re, W, and Ta possess a higher probability of cross-slip process than the alloying addition of Ti and therefore, these elements are predicted to be more efficient in strengthening the Ni_3_Al phase, which is worthy of further experimental evaluation and investigation. In addition, further analysis of the deformed samples will be carried out to elucidate the relationship between the compositions of the L1_2_-ordered phase and the underlying deformation mechanism.

## 4. Conclusions

Three multicomponent L1_2_-ordered single phase alloys were designed and fabricated to investigate the effect of multicomponent alloying conditions on the yielding behavior from 500 °C to 1000 °C. The primary conclusions can be summarized as follows.

Multicomponent L1_2_-ordered alloys, designed using the Calphad Thermo-Calc© software, were successfully fabricated via VAM and optimized heat treatment. Both microstructure and crystal structure analysis confirmed the formation of L1_2_-ordered single phase, demonstrating that ThermoCalc is a powerful software for intermetallic alloy design;The multicomponent L1_2_-ordered alloys exhibited a positive temperature dependence. The addition of Ti or Nb significantly increased strength up to the peak temperature;The addition of Ti or Nb enhanced the solid solution strengthening, ordering strengthening, and cross-slip-induced strengthening. The cross-slip-induced strengthening was the most dominant strengthening mechanism in L1_2_-ordered alloys and Ti or Nb addition remarkably increased the contribution of cross-slip-induced strengthening.

## Figures and Tables

**Figure 1 materials-17-02280-f001:**
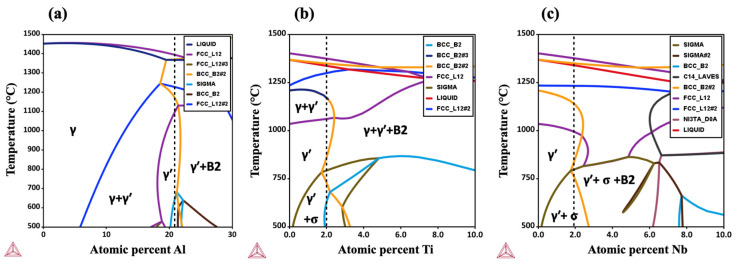
The pseudo-binary isopleth of three different multicomponent systems: (**a**) NiCo10Cr5Al, (**b**) NiCo10Cr5Al20Ti, and (**c**) NiCo10Cr5Al20Nb.

**Figure 2 materials-17-02280-f002:**
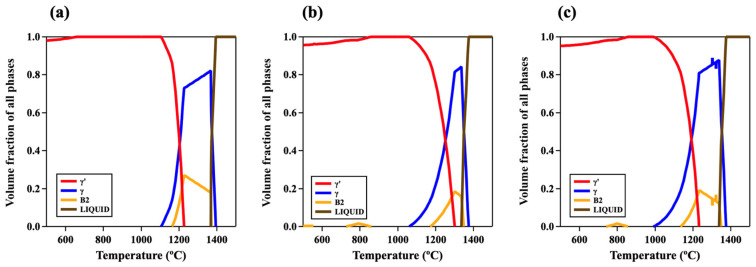
Volume fraction diagrams of the following multicomponent L1_2_-ordered intermetallic compounds: (**a**) CY1 (NiCoCrAl), (**b**) CY2 (NiCoCrAlTi), and (**c**) CY3 (NiCoCrAlNb).

**Figure 3 materials-17-02280-f003:**
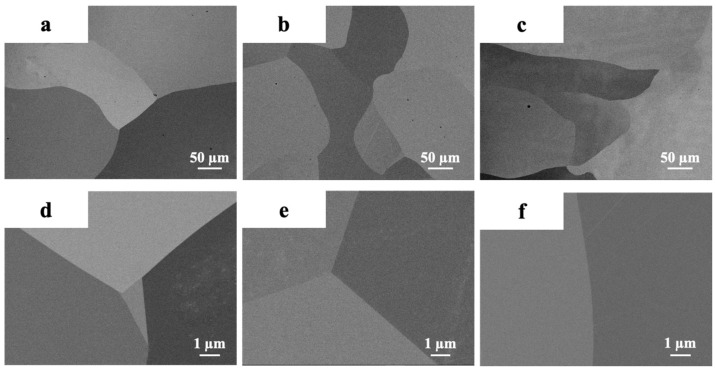
Backscatter images of microstructures after the heat treatments under different magnifications: (**a**) CY1 (250×), (**b**) CY2 (250×), (**c**) CY3 (250×), (**d**) CY1 (10,000×), (**e**) CY2 (10,000×), and (**f**) CY3 (10,000×).

**Figure 4 materials-17-02280-f004:**
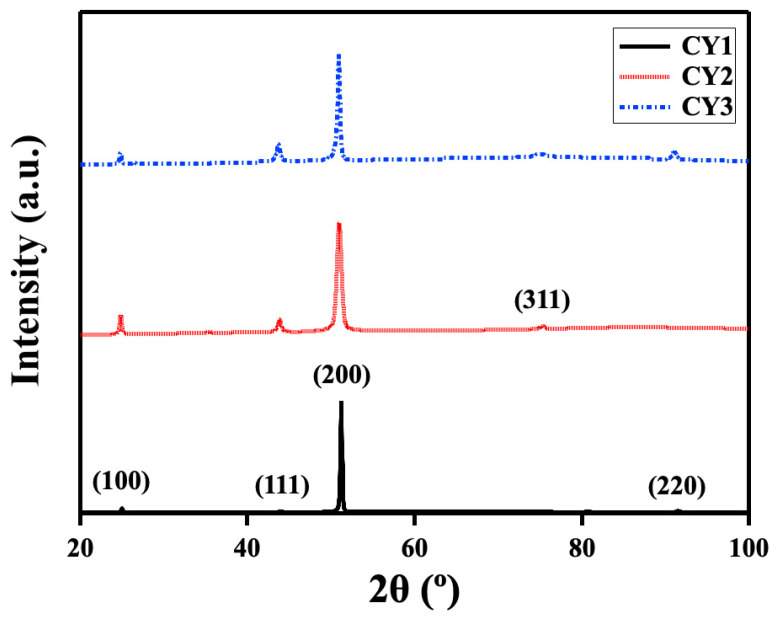
XRD analysis of the multicomponent L1_2_-ordered intermetallic alloys.

**Figure 5 materials-17-02280-f005:**
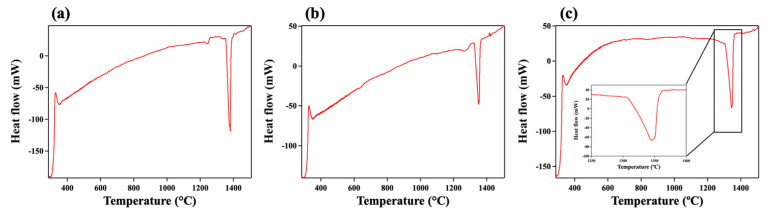
DTA heating curves of the studied alloys: (**a**) CY1, (**b**) CY2, and (**c**) CY3.

**Figure 6 materials-17-02280-f006:**
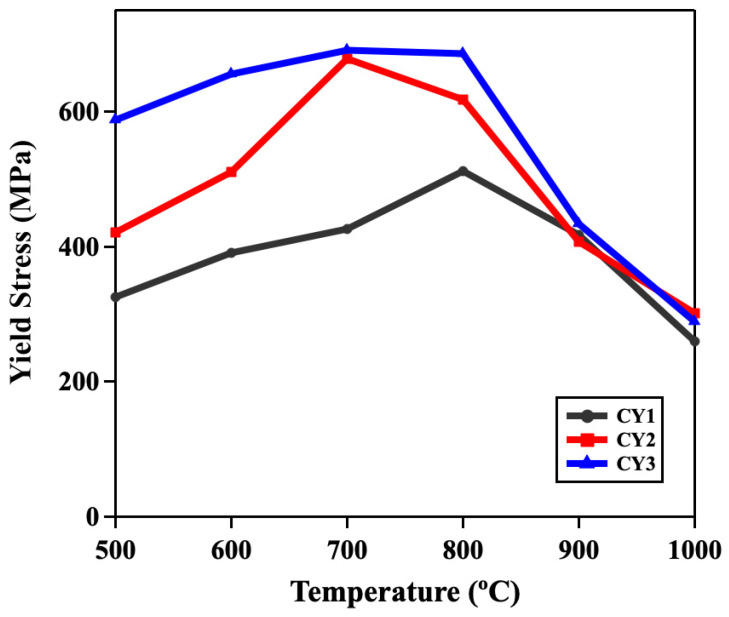
Temperature dependence of strengths.

**Figure 7 materials-17-02280-f007:**
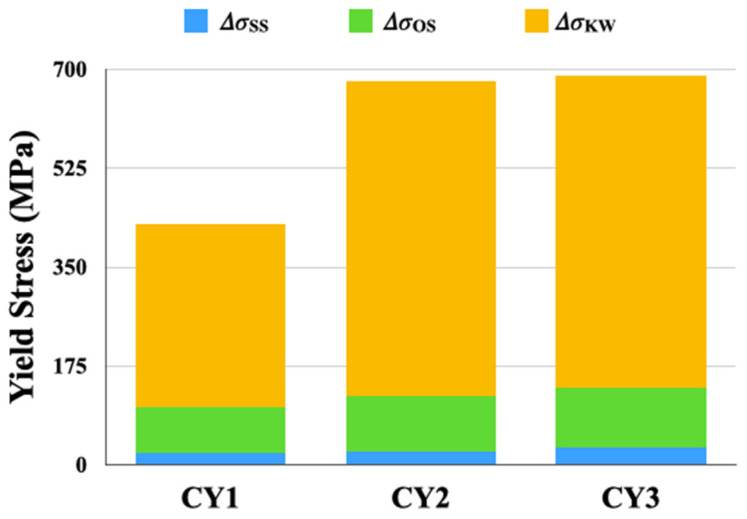
Strength contribution of different strengthening mechanisms.

**Table 1 materials-17-02280-t001:** Nominal compositions of the studied alloys (at%).

	Al	Co	Cr	Ni	Ti	Nb
CY1	21	10	5	64	-	-
CY2	20	10	5	63	2	-
CY3	20	10	5	63	-	2

**Table 2 materials-17-02280-t002:** The 2θ (°) values with related diffraction plane of L1_2_-ordered phase and the corresponding values in JCPDS files and the lattice constants of the studied alloys.

	(100)	(111)	(200)	(220)	(311)	Lattice Constant (Å)
JCPDS	24.90	43.60	50.70	75.02	91.21	-
CY1	24.98	43.87	51.51	75.02	91.50	3.48
CY2	24.78	43.98	50.59	74.81	91.14	3.58
CY3	24.77	44.03	51.00	75.24	91.18	3.59

**Table 3 materials-17-02280-t003:** Phase transformation temperatures and solidus temperatures for the studied alloys.

	CY1 (°C)	CY2 (°C)	CY3 (°C)
Phase transformation temperature	1217	1244	1298
Solidus temperature	1353	1324	1305

**Table 4 materials-17-02280-t004:** Yield strengths of the studied alloys at different temperatures.

(°C)	CY1 (MPa)	CY2 (MPa)	CY3 (MPa)
500	324.7	420.5	587.7
600	391.2	510.6	655.9
700	426.5	677.7	690.4
800	511.5	617.6	685.2
900	417.1	407.0	433.7
1000	259.5	300.9	288.7

**Table 5 materials-17-02280-t005:** Coefficients for change in APBE [22].

Coefficient	Co	Cr	Ti	Nb
*k_i_* (mJ m^−2^/at.pct)	−1.50	−5.00	15.00	21.40

## Data Availability

Data are contained within the article.

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
