# Peer review of "Effect of Alloying Elements on the High-Temperature Yielding Behavior of Multicomponent γ′-L12 Alloys"

_materials, 2024, doi:10.3390/ma17102280_

Round 1

Reviewer 1 Report

Comments and Suggestions for Authors

The article presents the results of research on the influence of Ti and Nb addition on high temperature yielding of g'-L1 alloys.

Below are my comments on the article.

The Introduction sufficiently describes the current state of knowledge with reference to the References and justifies undertaking the research topic.

The materials and methodology of the research were sufficiently described. However, I do have a point here. The authors analyze and select the chemical composition of alloys using at. %, while in the diagrams it is mole percent. This should be standardized. Moreover, I believe that it would be better to use wt.% because the authors describe the smelting of these alloys by giving the mass in [g].

Moreover, in Chapter 2 the authors refer to Figure 1a,b,c, while Figures 1d,e,f are also included, to which they refer only in Chapter 3.1. This is not an error, but it does introduce some disorganization of the text.

Results

I think chapter 3.1 has an inappropriate title. In this chapter, the authors focus mainly on the structure of these alloys, conducting a broader analysis on the transformation temperature. Therefore, this should not be understood as a property of alloys. I think a title on microstructure would be better.

Phase and alloy designations should be unified throughout the article.

Authors should revise the descriptions in tables. For example, I refer to Table 4. The first column shows Temperature. Therefore, the first cell of this column should have the description Temperature [oC] and not [MPa]. The unit [MPa] applies to columns 2, 3 and 4. The descriptions should be arranged in all tables.

The article presents the SEM microstructure (Figure 2). EDS analysis was not performed, therefore providing this information in Chapter 2 is unnecessary. Additionally, it is better to state that the microstructure tests were performed using BSE than to state that the microscope was equipped with such a detector. In general, this is a standard capability of a scanning microscope, and additionally, such a microscope has much wider capabilities. So there is no reason to expose this information.

Author Response

Reviewer #1

  1. The materials and methodology of the research were sufficiently described. However, I do have a point here. The authors analyze and select the chemical composition of alloys using at. %, while in the diagrams it is mole percent. This should be standardized. Moreover, I believe that it would be better to use wt.% because the authors describe the smelting of these alloys by giving the mass in [g].

Thank you very much for the comments and suggestions. The description of the pseudo-binary diagrams has been revised using at.%. As this research is related to intermetallic alloys, systematic alloy design is assumed to be better understood using the expression of at.%. (Figure 1)

  1. Moreover, in Chapter 2 the authors refer to Figure 1a,b,c, while Figures 1d,e,f are also included, to which they refer only in Chapter 3.1. This is not an error, but it does introduce some disorganization of the text.

Thank you very much for the comments and suggestions. The previous Figure 1 has been divided into two figures of Figure 1 and Figure 2 to improve the readability. (Figure 1, and Figure 2)

  1. I think chapter 3.1 has an inappropriate title. In this chapter, the authors focus mainly on the structure of these alloys, conducting a broader analysis on the transformation temperature. Therefore, this should not be understood as a property of alloys. I think a title on microstructure would be better.

Thank you very much for the comments and suggestions. Chapter 3.1's title has been changed to “Microstructures.” (Line 156)

  1. Phase and alloy designations should be unified throughout the article.

Thank you very much for the comments and suggestions. The phase and alloy designations have been unified to improve the readability. g’ is designated as g’ (L12-ordered) phase.

  1. Authors should revise the descriptions in tables. For example, I refer to Table 4. The first column shows Temperature. Therefore, the first cell of this column should have the description Temperature [oC] and not [MPa]. The unit [MPa] applies to columns 2, 3 and 4. The descriptions should be arranged in all tables.

Thank you very much for the comments and suggestions. The descriptions in the tables have been rearranged to improve the clarification. (Table 1, Table 3, and Table 4)

  1. The article presents the SEM microstructure (Figure 2). EDS analysis was not performed, therefore providing this information in Chapter 2 is unnecessary. Additionally, it is better to state that the microstructure tests were performed using BSE than to state that the microscope was equipped with such a detector. In general, this is a standard capability of a scanning microscope, and additionally, such a microscope has much wider capabilities. So there is no reason to expose this information.

Thank you very much for the comments and suggestions. The description of microstructure images has been revised using BSE. Unnecessary information regarding SEM has been removed from the context in line 140-141 in page 3. “The microstructures of each alloy were characterized using JEOL JSM-7200F scanning electron microscopy (SEM).”

Reviewer 2 Report

Comments and Suggestions for Authors

The authors have written about the effects of alloying elements on the high temp yielding of multicomponent alloys - high entropy systems. The tested various additions and revealed cross-slip induced strengthening as a key contributor to high temperature strength.

The study was performed with reliance primarily on the CALPHAD method. This was used to attribute experimentally observed performance to mechanisms of high temperature strengthening.

The paper is well written, informative and combines multiple methods to provide insights.

What is lacking in the paper is context and significance.

Specifically the authors need to review a broader range of literature and shed light on mechanisms observed in similar materials. Importantly, the peformance should be interpreted with reference to other materials.

authors must make the following three major changes to the work:

1. Which applications are these materials intended for? What are the current materials used in these applications now. Discuss the costs of presently used materials and attempt to quantify the benefits posed by these new materials under high temperature loading conditoins

2. How does the mechanical performance compare with superalloys at similar temperature and loading conditions?

3. The mechanisms here involve dislocation interactions. Does some annealing phenomenon at high temperature negate the strengthening mechsnism over time? How does the reported effect of grain-size (Hall Petch) compare with discussions and explanations from comprable materials in the literature? What does this mean for the processing of the alloy here?

4. The discussion is missing - discuss the significance of the results and how they compare with what is known about materials for such applications.

5. Improvements to the English are required - check the agreement of words in the sentences

Comments on the Quality of English Language

It is not badly written. However, a thorough grammar and syntax check is required 

Author Response

Reviewer #2

  1. Which applications are these materials intended for? What are the current materials used in these applications now. Discuss the costs of presently used materials and attempt to quantify the benefits posed by these new materials under high temperature loading conditoins

Thank you very much for the constructive comments and suggestions. We have revised the introduction section and added the applications to the revised manuscript. The main applications are structural materials used in aerospace engineering and the power generation industry. This was shown in the introduction (Line 37).

The costs of this material are comparable to commercial Ni-base superalloys because the constituent elements are the same. This research is a very basic study, and the materials are all single-phase to understand deformation behavior clearly. Generally, single-phase alloys are not used in application due to the lack of valance of several mechanical properties. In the future, our results will be useful to design new alloys with two-phase alloys. However, it is difficult to say the benefits of the tested alloys at the moment. Then, we just show the application of these materials.

  1. How does the mechanical performance compare with superalloys at similar temperature and loading conditions?

Thank you very much for the constructive comments. Ni-based superalloys generally possess ?+?ʹ dual-phase microstructures, and several strengthening mechanisms related to the interaction between the two phases, such as coherency strengthening and modulus mismatch strengthening, can occur. However, this research aims to understand the yielding behavior of multicomponent ?ʹ single-phase alloys. Therefore, we compared our investigated alloys with single-phase binary Ni3Al alloys and ternary Ni3Al alloys with the alloying addition of Ti or Nb instead of comparing them with ?+?ʹ dual-phase superalloys in the result and discussion section.

  1. The mechanisms here involve dislocation interactions. Does some annealing phenomenon at high temperature negate the strengthening mechanism over time? How does the reported effect of grain-size (Hall Petch) compare with discussions and explanations from comparable materials in the literature? What does this mean for the processing of the alloy here?

Thank you very much for the constructive comments. Before the heat treatments, there was no prior cold working process. The processing of heat treatments mainly involves obtaining single-phase microstructures. So, annealing has no effect on dislocation density.

It is generally accepted that grain sizes under the ranges of 100 – 1000 μm do not produce significant strength enhancements in Ni3Al alloys, and the level of grain boundary strengthening decreases with increasing temperature; the effect of grain size on high temperature strength is limited.

In this study, we want to have a single L12 phase alloy. Then, after we designed the alloy composition, we just used the conventional arc melting method to prepare ingots. If we can get a single phase, any processing method is fine.

  1. The discussion is missing - discuss the significance of the results and how they compare with what is known about materials for such applications.

Thank you very much for the constructive comments and suggestions. We added the discussion as shown in the follows. (Line 338-353)

From a technical point of view, the peak temperature is usually at the highest temperature where the materials are used [44]. Ni-based superalloys [45-48] and L12 strengthened HEAs [49-52] have exhibited anomalous positive temperature dependence of strength at intermediate temperatures followed by a decrease in yield strength with increasing temperatures. The decrease in strength has been explained mainly by degradation of microstructures and transition of deformation mechanisms. Since the L12-ordered single phase microstructure in our systematically designed alloys was thermodynamically stable at the peak temperatures, the possibility of a microstructure degradation has been ruled out for the cause of the decreased yield strength above the peak temperatures. Our results revealed the intrinsic capability of the multicomponent L12-ordered phases and suggested that the difference in the yield strength before the peak temperatures was significantly influenced by the difference in the contribution of cross-slip-induced strengthening, which was strongly associated with the alloying compositions of the multicomponent L12-ordered phase. This work has shed light on the significance of cross-slip-induced strengthening in multicomponent L12-ordered phase and can serve as a guideline for future design of L12 strengthened HEAs.

  1. Improvements to the English are required - check the agreement of words in the sentences

Thank you very much for the comments and suggestions. We have used two kinds of online writing assistance to revise the entire manuscript to meet the standard of English for publication.

Reviewer 3 Report

Comments and Suggestions for Authors

The paper Effect of alloying elements on the high temperature yielding behavior of multicomponent ʹ-L1 2 alloys is in the subject  Materials MDPI journal.

The paper presents the high temperature compression test from 500 °C to 1000 °C revealed a positive temperature dependence of strength before reaching the peak strength in the studied alloys: NiCoCrAl, NiCoCrAlTi, and NiCoCrAlNb. Ti and Nb alloying addition significantly enhanced the high temperature yield strengths before the peak temperature. An article presented well planned experiments. The results are clearly presented and analysed.

An article needs minor review to improve found mistakes. In review please consider my remark presented in a few comments below

Comment 1

I propose adding of article structure at the end of the Introduction to improve article readability

Comment 2

In Section 2 you wrote

We developed three systems: (a) NiCoCrAl, (b) NiCoCrAlTi, and (c)NiCoCrAlNb.

I propose using of bullet list

We developed three systems:

·         NiCoCrAl,

·         NiCoCrAlTi,

·         NiCoCrAlNb.

Comment 2

You present

Figure 1. T he pseudo-binary isopleth of three different multicomponent systems: (a) NiCo10Cr5Al, (b) NiCo10Cr5Al20Ti, and (c) NiCo10Cr5Al20Nb and the volume fraction diagrams of the multicomponent L12-ordered intermetallic compounds: (d) CY1 (NiCoCrAl), (e) CY2 (NiCoCrAlTi), and (f) CY3 (NiCoCrAlNb)

In Table 1

Table 1. Nominal compositions of the studied alloys at% Al Co Cr Ni Ti Nb

You consider CY1, Cy2 and CY3 components

Please considering divide this figure on two

Figure 1. The pseudo-binary isopleth of three different multicomponent systems: (a) NiCo10Cr5Al, (b) NiCo10Cr5Al20Ti, and (c) NiCo10Cr5Al20Nb

Figure 2. Volume fraction diagrams of the multicomponent L12-ordered intermetallic compounds: (a) CY1 (NiCoCrAl), (b) CY2 (NiCoCrAlTi), and (c) CY3 (NiCoCrAlNb)

Comment 3

You wrote Section 3

3. Results

Because you present both results and its discussion. I propose changing of name  Section 3 on

3. Results and Discussion

Comment 4

I propose  decreasing the magnification of

Figure 5. Temperature dependence of strengths

Comment 5

You wrote relevant Conclusions

Three multicomponent L1 2 single phase alloys were designed and fabricated to investigate the effect of multicomponent alloying conditions on the yielding behavior from 500 °C to 1000 °C. The primary conclusions can be summarized as follows.

(1) Multicomponent L1 2 alloys designed using the Calphad Thermo-Calc© software were successfully fabricated via VAM and optimized heat treatment. Both microstructure and crystal structure analysis confirmed the formation of L1 2 single phase, demonstrating that ThermoCalc is a powerful software for intermetallic alloy design.

(2) The multicomponent L1 2 alloys exhibited a positive temperature dependence. The addition of Ti or Nb significantly increased in strength up to the peak temperature.

(3) The addition of Ti or Nb enhanced the solid solution strengthening, ordering strengthening, and cross-slip-induced strengthening. Cross-slip-induced strengthening was the most dominant strengthening mechanism in L12 alloys, and Ti or Nb addition remarkably increased the contribution of cross-slip-induced strengthening

Because numbering was used for Sections please consider using the bullet list

Three multicomponent L1 2 single phase alloys were designed and fabricated to investigate the effect of multicomponent alloying conditions on the yielding behavior from 500 °C to 1000 °C. The primary conclusions can be summarized as follows:.

·         Multicomponent L1 2 alloys designed using the Calphad Thermo-Calc© software were successfully fabricated via VAM and optimized heat treatment. Both microstructure and crystal structure analysis confirmed the formation of L1 2 single phase, demonstrating that ThermoCalc is a powerful software for intermetallic alloy design.

·         The multicomponent L1 2 alloys exhibited a positive temperature dependence. The addition of Ti or Nb significantly increased in strength up to the peak temperature.

·         The addition of Ti or Nb enhanced the solid solution strengthening, ordering strengthening, and cross-slip-induced strengthening. Cross-slip-induced strengthening was the most dominant strengthening mechanism in L12 alloys, and Ti or Nb addition remarkably increased the contribution of cross-slip-induced strengthening.

Comment 6

Can you add any suggestions the future research or implementation to the Conclusions?

Author Response

Reviewer #3

  1. I propose adding of article structure at the end of the Introduction to improve article readability

Thank you very much for the comments and suggestions. The final paragraph of the introduction now includes a description of the article structure, including the alloy design, the evaluation of yielding behaviors, and the analysis of strengthening mechanisms. (Line 114-117)

“This work systematically varied compositions by substituting Ni and Al with Ti or Nb to study the compositional effects on the yielding behaviors of the L12 phase. The investigated multicomponent L12 alloys were systematically designed using the thermodynamics software CALPHAD Thermo-Calc© [30]. The yielding behaviors of the multicomponent L12 alloys were evaluated by high temperature compression tests. The analysis and discussion will delve into the contribution of different strengthening mechanisms under various alloying conditions.”

  1. In Section 2 you wrote, We developed three systems: (a) NiCoCrAl, (b) NiCoCrAlTi, and (c)NiCoCrAlNb. I propose using of bullet list We developed three systems:
  •  NiCoCrAl,
  •  NiCoCrAlTi,
  •  NiCoCrAlNb.

Thank you very much for the suggestions. The type of bullet list has been applied to improve the readability. (Line 125-127)

  1. You present

Figure 1. The pseudo-binary isopleth of three different multicomponent systems: (a) NiCo10Cr5Al, (b) NiCo10Cr5Al20Ti, and (c) NiCo10Cr5Al20Nb and the volume fraction diagrams of the multicomponent L12- ordered intermetallic compounds: (d) CY1 (NiCoCrAl), (e) CY2 (NiCoCrAlTi), and (f) CY3 (NiCoCrAlNb)

In Table 1

Table 1. Nominal compositions of the studied alloys at% Al Co Cr Ni Ti Nb

You consider CY1, Cy2 and CY3 components Please considering divide this figure on two

Figure 1. The pseudo-binary isopleth of three different multicomponent systems: (a) NiCo10Cr5Al, (b) NiCo10Cr5Al20Ti, and (c) NiCo10Cr5Al20Nb

Figure 2. Volume fraction diagrams of the multicomponent L12-ordered intermetallic compounds: (a) CY1 (NiCoCrAl), (b) CY2 (NiCoCrAlTi), and (c) CY3 (NiCoCrAlNb)

Thank you very much for the suggestions. The previous Figure 1 has been divided into two figures to improve the readability.

  1. I propose decreasing the magnification of Figure 5. Temperature dependence of strengths

Thank you very much for the suggestions. The Figure 5 has been rearranged with decreased magnification.

  1. You wrote relevant Conclusions

Three multicomponent L1 2 single phase alloys were designed and fabricated to investigate the effect of multicomponent alloying conditions on the yielding behavior from 500 °C to 1000 °C. The primary conclusions can be summarized as follows.

(1) Multicomponent L1 2 alloys designed using the Calphad Thermo-Calc© software were successfully fabricated via VAM and optimized heat treatment. Both microstructure and crystal structure analysis confirmed the formation of L1 2 single phase, demonstrating that ThermoCalc is a powerful software for intermetallic alloy design.

(2) The multicomponent L1 2 alloys exhibited a positive temperature dependence. The addition of Ti or Nb significantly increased in strength up to the peak temperature.

(3) The addition of Ti or Nb enhanced the solid solution strengthening, ordering strengthening, and cross-slip- induced strengthening. Cross-slip-induced strengthening was the most dominant strengthening mechanism in L12 alloys, and Ti or Nb addition remarkably increased the contribution of cross-slip-induced strengthening

Because numbering was used for Sections please consider using the bullet list

Three multicomponent L1 2 single phase alloys were designed and fabricated to investigate the effect of multicomponent alloying conditions on the yielding behavior from 500 °C to 1000 °C. The primary conclusions can be summarized as follows:.

  • Multicomponent L1 2 alloys designed using the Calphad Thermo-Calc© software were successfully fabricated via VAM and optimized heat treatment. Both microstructure and crystal structure analysis confirmed the formation of L1 2 single phase, demonstrating that ThermoCalc is a powerful software for intermetallic alloy design.
  • The multicomponent L1 2 alloys exhibited a positive temperature dependence. The addition of Ti or Nb significantly increased in strength up to the peak temperature.
  • The addition of Ti or Nb enhanced the solid solution strengthening, ordering strengthening, and cross-slip- induced strengthening. Cross-slip-induced strengthening was the most dominant strengthening mechanism in L12 alloys, and Ti or Nb addition remarkably increased the contribution of cross-slip-induced strengthening.

Thank you very much for the suggestions. The bullet list type has been applied to avoid misunderstandings. (Line 365-377)

  1. Can you add any suggestions for future research or implementation to the Conclusions?

Thank you very much for the constructive suggestions. According to previous literature, Re, W, and Ta also possess a high probability of a cross-slip process, and therefore, these elements are predicted to be efficient in strengthening the Ni3Al phase. Future research on multicomponent alloys with the addition of Re, W, and Ta could be performed to reveal their capability of cross-slip-induced strengthening and compare them to the results in this research. (Line 354-358)

Reviewer 4 Report

Comments and Suggestions for Authors

Authors presented results of an interesting research on alloying elements Titanium and Niobium influence on the high temperature yielding behavior of NiCoCrAl alloys. Those results are impressive and quite convincing. There is not much to be added to the presented conclusions.

- My suggestion is to add, if possible, a discussion, which other alloying elements could produce the same results on increasing the yield strength of the tested alloys, besides the two that you applied. Could that be the direction for the further research on this phenomenon?

- The remarks on presentation (style-wise) are only concerning small problems with placing of determinate articles. Otherwise, the level of English language is quite good.

- Another remark concerns the List of references, at the end of the article, which is not written in accordance with the requirements of the Journal’s template.

 The scanned pages of the manuscript, with marked errors and suggested corrections are enclosed.

Author Response

Reviewer#4

  1. My suggestion is to add, if possible, a discussion, which other alloying elements could produce the same results on increasing the yield strength of the tested alloys, besides the two that you applied. Could that be the direction for the further research on this phenomenon?

Thank you very much for the constructive suggestions. Direction for the future research on this phenomenon has been added to the final paragraph in the result and discussion section. According to previous literature, Re, W, and Ta also possess a high probability of a cross-slip process, and therefore, these elements are predicted to be efficient in strengthening the Ni3Al phase. Future research on multicomponent alloys with the addition of Re, W, and Ta could be performed to reveal their capability of cross-slip-induced strengthening and compare them to the results in this research. (Line 354-358)

  1. The remarks on presentation (style-wise) are only concerning small problems with placing of determinate articles. Otherwise, the level of English language is quite good.

Thank you very much for the constructive comments and suggestions. We have revised the manuscript according to your valuable peer-review file and two kinds of online writing assistance to meet the standard of English for publication.

  1. Another remark concerns the List of references, at the end of the article, which is not written in accordance with the requirements of the Journal’s template.

Thank you very much for pointing out the mistake. The List of References has been revised to comply with MDPI's requirements.
